# Deep Learning-Based Periapical Lesion Detection on Panoramic Radiographs

**DOI:** 10.3390/diagnostics15040510

**Published:** 2025-02-19

**Authors:** Viktor Szabó, Kaan Orhan, Csaba Dobó-Nagy, Dániel Sándor Veres, David Manulis, Matvey Ezhov, Alex Sanders, Bence Tamás Szabó

**Affiliations:** 1Department of Oral Diagnostics, Faculty of Dentistry, Semmelweis University, 47 Szentkiralyi Str., 1088 Budapest, Hungary; knorhan@dentistry.ankara.edu.tr (K.O.); dobo-nagy.csaba@semmelweis.hu (C.D.-N.); szabo.bence.tamas@semmelweis.hu (B.T.S.); 2Department of Dentomaxillofacial Radiology, Faculty of Dentistry, Ankara University, Ankara 06100, Turkey; 3Medical Design Application, and Research Center (MEDITAM), Ankara University, Ankara 06100, Turkey; 4Department of Biophysics and Radiation Biology, Semmelweis University, 1088 Budapest, Hungary; daniel.s.veres@gmail.com; 5Diagnocat Inc., San Francisco, CA 94102, USA; david@diagnocat.com (D.M.); matvey@diagnocat.com (M.E.); alex@diagnocat.com (A.S.)

**Keywords:** artificial intelligence, deep learning, dental digital radiography, panoramic radiography, periapical diseases

## Abstract

**Background/Objectives**: Our study aimed to determine the accuracy of the artificial intelligence-based Diagnocat system (DC) in detecting periapical lesions (PL) on panoramic radiographs (PRs). **Methods:** 616 teeth were selected from 357 panoramic radiographs, including 308 teeth with clearly visible periapical radiolucency and 308 without any periapical lesion. Three groups were generated: teeth with radiographic signs of caries (Group 1), teeth with coronal restoration (Group 2), and teeth with root canal filling (Group 3). The PRs were uploaded to the Diagnocat system for evaluation. The performance of the convolutional neural network in detecting PLs was assessed by its sensitivity, specificity, and positive and negative predictive values, as well as the diagnostic accuracy value. We investigated the possible effect of the palatoglossal air space (PGAS) on the evaluation of the AI tool. **Results:** DC identified periapical lesions in 240 (77.9%) cases out of the 308 teeth with PL and detected no PL in 68 (22.1%) teeth with PL. The AI-based system detected no PL in any of the groups without PL. The overall sensitivity, specificity, and diagnostic accuracy of DC were 0.78, 1.00, and 0.89, respectively. Considering these parameters for each group, Group 2 showed the highest values at 0.84, 1.00, and 0.95, respectively. Fisher’s Exact test showed that PGAS does not significantly affect (*p* = 1) the detection of PL in the upper teeth. The AI-based system showed lower probability values for detecting PL in the case of central incisors, wisdom teeth, and canines. The sensitivity and diagnostic accuracy of DC for detecting PL on canines showed lower values at 0.27 and 0.64, respectively. **Conclusions:** The CNN-based Diagnocat system can support the diagnosis of PL on PRs and serves as a decision-support tool during radiographic assessments.

## 1. Introduction

One of the most important additional diagnostic methods is radiological examination in dentistry, which can aid in the detection of periapical lesions (PLs) [1]. PL develops along the root of the tooth, usually preceded by the necrosis of the pulp. It is caused by bacteria penetrating through the root canal into the periapical space, resulting in a progressive process inducing bone resorption with a characteristic radiographic appearance [2,3]. Based on the latest systematic review and meta-analysis, Tibúrcio-Machado et al. [4] found a 52% prevalence of PL in adults, and current clinical practice considers the use of radiological imaging to be of crucial importance for a definitive diagnosis [5]. Various radiographic methods such as periapical and panoramic radiographs (PRs), and additionally, cone beam computed tomography (CBCT), are some of the most commonly used modalities for the detection of apical lesions [6]. Even though CBCT exhibits notably superior discriminatory capabilities compared to periapical radiographs [7], its widespread usage is limited due to higher patient doses and significant associated costs, making it applicable to only a few specific indications. PRs, which are widely used in dental clinics, can provide the dentist with numerous data concerning diagnosis and treatment planning [8]. Their widespread use may be justified by the fact that they provide the dentist with a relatively good overview of the patient’s dental status with a relatively lower radiation dose. Noteworthily, there is a substantial difference among dental professionals in the reading of PRs based on their skills and previous experience [9]. On the other hand, the image quality and accuracy of the radiographic evaluation of PRs are greatly influenced by position errors such as the incorrect position of the tongue during exposure. The presence of a radiolucent air space between the dorsal surface of the tongue and the palate (palatoglossal air space, PGAS) can impair the evaluation of the apical region of the maxillary teeth, which can lead to misdiagnoses of PL [10]. Periapical radiographs are the most frequently used imaging technique for identifying changes in the periapical region [11]. However, both periapical and PRs face inherent challenges, including the superimposition of anatomical structures, geometrical distortion, anatomical noise, and the limitation of being two-dimensional in nature [12].

Computer-aided diagnosis (CAD) software supports clinicians during their evaluation. These software draw the attention of clinicians to areas with potential pathologies on medical images, and thus can significantly enhance and expedite the work, providing valuable support in their daily tasks [13]. Using artificial intelligence (AI)-based systems in dental radiology can provide many advantages and increase the accuracy and speed of diagnostic assessment [14,15]. Radiology was one of the first medical fields where AI appeared. This is primarily due to the generation of an enormous amount of digital radiographic data, which serves as the foundation for the development and application of AI, and the larger the amount and heterogeneity of data it encounters, the more accurately it can solve a given task [16]. Deep learning (DL) as a subset of AI learns and improves without specific instructions or programming [17]. In radiology, a subgroup of neural networks, convolutional neural networks (CNNs), can be used. U-Net is a widely applied CNN architecture in image analysis, which was created to efficiently leverage a limited dataset while upholding both speed and accuracy [18]. In the field of dentistry, DL can be used for numerous tasks including the detection and classification of caries lesions on PRs [19], the segmentation and detection of PLs on CBCT images [20], the detection of periodontal bone loss on PRs [21], and cephalometric landmark detection [22].

Currently, there is still growing interest in the AI-assisted detection of PL on PRs [6,23,24,25,26,27]. This research aims to evaluate the effectiveness of an AI-based tool in a real clinical setting, focusing on its role in assisting with radiographic evaluations of PL on PRs. This study focused on key metrics such as sensitivity, specificity, predictive values (both positive and negative), and the proportion of correct classifications, using the ground truth provided by experienced dentomaxillofacial radiologists.

## 2. Materials and Methods

In our retrospective radiographic research, 357 PRs were collected and assessed by three independent observers: a fifth-year dental student, a general dentist, and a dentomaxillofacial radiologist with more than ten years of experience (Figure 1). Before the evaluation of PRs, observers underwent calibration via a training session supervised by a senior dentomaxillofacial radiologist with more than thirty years of experience. They reviewed standardized radiographs and discussed possible pitfalls during the assessment of PRs. Regarding the PRs, 308 teeth with visible periapical radiolucency and 308 teeth without any periapical radiolucency (PARL) were selected. During the selection of teeth free of PARL, the tooth type was matched with the tooth type of teeth with periapical lesions. All the selected teeth had to meet one of the following additional inclusion criteria: a carious lesion was visible on the dental crown (Group 1), restoration in the dental crown was present without root canal filling (Group 2), or a root canal filling was present (Group 3). Furthermore, the presence of palatoglossal airspace (PGAS) was recorded if it was visible on the apical region of the selected upper tooth. Endo-periodontal lesions and obvious radiographic signs of previous apical surgery were considered exclusion criteria. The ground truth was determined during personal sessions by two dentomaxillofacial radiologists, one with more than ten and one with more than thirty years of experience. In instances where there was disagreement regarding the detection of PL on the selected PRs, a consensus was achieved in every case.

All radiographs were acquired at the Department of Oral Diagnostics, Faculty of Dentistry, Semmelweis University using an Orthopantomograph 3D Pro appliance (KaVo, Biberach an der Riss, Germany) with the following settings: 66 kV, 5.0 mA, and 16 s. The selection was performed using the department’s patient archiving and communication system (IMPAX software, v.6.5.2.657, Agfa HealthCare, Mortsel, Belgium). The human observers were allowed to adjust the brightness or contrast and were able to use magnification in the IMPAX software. The PRs were evaluated on a Samsung S24F350FHU (Samsung, Seoul, Republic of Korea) (full HD, resolution: 1920 × 1200 pixels) monitor. The metadata for the selected radiographs were documented, including the study date, patient age, and the specific tooth and its group depicted in the radiograph. This information was stored separately from the downloaded anonymized image data. The applied DC system was provided only by the encrypted anonymized image files. The study protocol was performed in accordance with the Declaration of Helsinki and approved by the Semmelweis University Regional and Institutional Committee of Science and Research Ethics (SE RKEB 138/2020). Access to the collected anonymized image files was restricted to observers, while only human observers had access to the metadata. It is important to note that the sample selection process did not favor any particular sex.

To visualize and understand the possible effect of projection geometry of upper and lower periapical regions [28], one of the authors (B.T.S.) of the present manuscript provided his functional cast and cone-beam computed tomography (CBCT) data acquired for other treatment purposes. Regarding the CBCT data, the distances between the radiographic apex of the upper and lower first molars, canines, central incisors, and the vestibular surface of the alveolar process were determined. In the case of multirooted teeth, the geometric center of apices was used for the distance measurement. Hereinafter, holes were prepared on the cast using a dental bur at the selected spots with the prerecorded depth values. Into each prepared cavity of the upper cast, a 5 mm diameter aluminum ball was inserted and fixed with dental wax. Subsequently, the upper and lower casts were fitted in the dental intercuspidal position with dental wax and a PR was taken using the same panoramic X-ray unit (Orthopantomograph 3D Pro). This process was also performed for the lower cast after transferring the aluminum balls from the upper sample. This approach was presumed to be useful in assessing how the standard settings of the panoramic imaging system used in our study affect the projection geometry of an object with uniform dimensions in all directions. By placing a known reference object (aluminum ball) in specific anatomical locations, we aimed to analyze potential distortions or magnifications introduced by the PR under these standard settings. This evaluation provides insights into how different anatomical regions are projected onto the final image and helps us better interpret the results obtained from panoramic radiographs in periapical diagnostics.

A 95% confidence interval (CI) of sensitivity, specificity, and other diagnostic test parameters was calculated using the exact binomial method. For teeth with PARL, additional statistical analyses were conducted. As a univariate analysis, we calculated the hit rate with its CI for all cases and separately for Group 1, Group 2, and Group 3. As we assumed that the hit rate of the AI prediction depends on the type of teeth, a logistic regression model was also applied. The outcome variable was the AI prediction, and Group 1, Group 2, and Group 3 were used as predictors to control the effect for the tooth type as additive terms in multivariate analyses. The possible correlation between the software evaluation and position error derived from PGAS was also assessed using Fisher’s exact test. Statistical analyses were performed using R software (v4.3.2) [29]. For diagnostic test parameters, Stevenson’s epiR (v2.0.66) was used; for the logistic regression models, Harrell’s rms (v6.7.1) was used; and for interpreting the regression results on the plot, Lüdecke’s sjPlot (v2.8.15) packages were used. Statistical significance was set at *p* < 0.05.

## 3. Results

From the 357 PRs, 308 teeth with clearly visible PARL and 308 teeth without PARL were selected. Human observers collected 100 (32.5%) teeth with carious lesions (Group 1), 102 (33.1%) teeth with restoration in the dental crown (Group 2), and 106 (34.4%) teeth containing root canal filling teeth (Group 3) with PARL and imported them into DC system for evaluation. The CNN found periapical lesions in 240 (77.9%) cases out of the 308 teeth with PARL: for 204 teeth, ’periapical radiolucency’ was reported, and for 36 teeth, ’root fragment’ was reported. However, the DC identified no PARL in 68 (22.1%) teeth with PARL. The AI-based system detected no PARL in any of the groups without PARL: 70 (22.7%) teeth with radiographic signs of caries, 228 (74.0%) with restoration, and 10 (3.2%) with root canal filling without PARL (Table 1 and Table 2).

The performance of the software was assessed based on its sensitivity, specificity, positive and negative predictive values, and correctly classified proportion values (Table 3). Noteworthily, the correctly classified proportion value can be considered the diagnostic accuracy since it is expressed as a proportion of correct predictions (TP + TN) among all predictions (TP + TN + FP + FN) [30,31]. The overall sensitivity, specificity, and diagnostic accuracy of DC were 0.78, 1.00, and 0.89, respectively. Considering these parameters for each group, Group 2 showed the highest values at 0.84, 1.00, and 0.95, respectively.

The results of the logistic regression models showed that the tooth type has a significant effect (*p* < 0.0001) on the prediction rate of DC. The AI-based system showed lower probability values for detecting PARL in the case of central incisors, wisdom teeth, and canines, with the latter showing the lowest values (Figure 2). In line with this result, the sensitivity and diagnostic accuracy of DC for detecting PARL on canines showed lower values at 0.27 and 0.64, respectively (Table 2).

Among teeth with PARL, 113 out of the 308 teeth were maxillary teeth. The DC system identified periapical lesions in 82 teeth, though 31 teeth with PARL remained undetected. Of the correctly diagnosed PRs, 50 contained PGAS (72.5%), while of the PRs with unidentified PARL, 19 contained PGAS (72.7%). Fisher’s Exact test showed that PGAS does not have a significant effect (*p* = 1) on the detection of PARL in the upper teeth.

The opacities of the aluminum balls embedded in the periapical area of the central incisor, canine, and the first molar showed no distortion on the PRs of the prepared casts (Figure 3).

## 4. Discussion

Our retrospective study aimed to determine the reliability of the AI-based DC system in predicting PARL on PRs as observed by human observers. Several other studies [6,23,25,26,32,33,34,35,36,37,38] show the scientific interest in detecting PL on different imaging modalities supported by CNN-based algorithms.

The model pipeline we applied was as follows: First, an ROI detection algorithm was trained specifically for teeth detection. The training dataset comprised around 4500 radiographs of teeth that were tagged, including instances of missing teeth. The models were designed to identify individual teeth, precisely outline their borders, and assign each tooth a unique identifier. The task was performed using a two-stage detector, Mask R-CNN, with a pretrained ResNet-101 backbone. The region of interest was defined based on the model’s object detection predictions, which included bounding boxes and segmentation masks, allowing for the assignment of numerical values to each tooth. To define the mouth region, the coordinates were determined by identifying the minimum and maximum x and y values of all detected teeth and then extending this area by a set number of pixels.

Initially, pseudo segmentations obtained from a pre-trained model were added to all panoramic photos. Following this, each image was cropped based on the predictions from the ROI detector. These cropped images were then input into the model for further processing [39,40].

The architecture selected for the model was Cascade R-CNN, which was trained on 5000 partially annotated PRs containing periapical lesions and acquired by various panoramic imaging units. In contrast to the Mask R-CNN model, Cascade R-CNN iteratively improves object recognition and segmentation by refining the bounding boxes. Image classification is performed by taking the average outcome for each cascade layer. This enhances the accuracy of forecasting and mitigates the problem of overfitting [41]. To enhance the model’s performance and generalization, various augmentations were applied to the input data. These included random cropping, rotation, brightness adjustments, contrast variations, downscaling, blurring, noise addition, optical and grid distortions, and contrast-limited adaptive histogram equalization (CLAHE) [36].

The PRs of the selected teeth were imported individually into the DC system for assessment. The DC recognized and numbered the teeth, and then highlighted the pathological lesions detected for each tooth. For the detected periapical lesion of the selected tooth, the term ‘Periapical radiolucency’ appeared under the name of the lesion detected by the DC. The anatomical localization of the PARL detected was indicated by the software with a green bounding box. Noteworthily, the DC assigns the term ‘Root fragment’ in progressive cases where carious lesions had almost destroyed the tooth crown. These cases were recorded manually and managed as the DC indicated the lesion. 

Li et al. [34] developed a CNN to detect caries and periapical periodontitis on periapical radiographs. They used 4129 periapical radiographs for training, validation, and testing of the ResNet-18-based CNN. The proposed DL model achieved sensitivity and specificity values of 0.82 and 0.84, respectively. The sensitivity and specificity values of the proposed AI model were comparable with our results. Additionally, Issa et al. [35] examined the accuracy of the U-Net-based DC system in the detection of PLs on periapical radiographs. A total of 60 teeth were evaluated by the DC, resulting in a sensitivity value of 0.92 and a specificity value of 0.97. Even though the sensitivity value was higher than what we obtained, the specificity value was in line with the present study. In their study, Pauwels et al. [37] aimed to compare the diagnostic performance of CNN with the performance of human observers for the detection of simulated periapical lesions on periapical radiographs. They created ten bone defects in bovine ribs to simulate PLs and acquired the periapical radiographs. The latter were evaluated by three oral radiologists and assessed by the proposed CNN. The human observers obtained a sensitivity value of 0.58, whilst the CNN showed a value of 0.79. The specificity values were similar in both cases, at 0.88 and 0.83, respectively. Although the sensitivity values were comparable in both studies, the specificity was higher in our study.

To the best of our knowledge, there is only a limited number of studies available assessing the performance of PL detection on PR radiographs. Celik et al. [23] examined ten different DL models’ performance in the detection of PL using 357 PRs. The highest sensitivity values were achieved using YOLOv3 and Dynamic R-CNN: 0.875 and 0.818, respectively. In a separate study, Ba-Hattab et al. [6] developed a DL architecture and tested it on 143 PRs comprising 299 PLs and reported overall sensitivity and specificity values of 0.722 and 0.856, respectively. These findings are comparable with our overall sensitivity and specificity values of 0.78 and 1.0, respectively. Notably, Ekert et al. [26] obtained a lower overall sensitivity value (0.65) using a deep learning algorithm based on six independent observers’ evaluations of PLs on 85 PRs, with the results of diagnostic performance reported by tooth type. The lowest sensitivity value of 0.52 was achieved for canines, similar to our study where the sensitivity was also the lowest among the tooth types (0.27). Additionally, PL detection of incisors and wisdom teeth showed lower probability values of the DC system. Thus, we considered it to be reasonable to examine the possible effect of PGAS and distortion of depicted periapical areas on PL detection. Nevertheless, based on the results of Fisher’s exact test (*p* = 1) applied for PGAS and the resulting distortion-free depiction of the aluminum balls on PRs, these factors might be excluded. In another study, Song et al. [25] used a U-Net-based CNN for the segmentation of PLs on PRs. The PLs were manually labeled by three oral and maxillofacial radiologists. They tested the software on 180 PLs, where the software segmented 147 PLs as PL and the sensitivity values were between 0.74 and 0.82, which are comparable with our results (0.78). Orhan et al. [36] used 100 PRs to assess the reliability of the DC system based on the diagnosis of various pathologic conditions, including PLs on PRs. During the ground truth evaluation, three oral and maxillofacial radiologists with different levels of clinical experience assessed the radiographs. Subsequently, PRs were imported into the DC system and the reliability was expressed in sensitivity and specificity values. In terms of PLs, the DC achieved a sensitivity value of 0.46, which is lower than our result. The specificity showed a value of 0.98 for the detection of PLs, which is in line with our findings (1.00).

Noteworthily, several studies were conducted reporting PL detection based on data of spatial imaging modalities. Fu et al. [33] proposed and validated the CNN-based PAL-Net algorithm for the detection of PLs on CBCT image data. They tested the software on one internal independent and three external datasets comprising 324, 393, 464, and 929 teeth, respectively. The CNN achieved sensitivities of 0.94, 0.90, and 0.88. Similarly, Orhan et al. [32] investigated the reliability of the DC system in the detection of PL on CBCT data. The U-Net-based system detected 142 PLs out of the 153 PLs, and a sensitivity value of 0.89 was reported. In both CBCT-based studies, the sensitivity values were higher than in our study (0.78), in which PRs were applied.

The aforementioned studies confirm the growing interest in AI-supported radiographic assessments. Despite deep learning’s advancements in detection, challenges persist for clinical integration. Studies require extensive, diverse datasets from various sources for reliability. The absence of standardized, public datasets hinders comparison and evaluation. The lack of a consistent evaluation standard makes it difficult to compare models effectively or determine optimal practices and strategies. AI-based tools still need continuous improvement to independently diagnose PLs on selected radiographs. Our study’s limitation lies in the exclusive reliance on visual diagnosis through radiograph inspection in panoramic radiographs (PRs). Achieving higher diagnostic accuracy requires integrating clinical data like percussion, thermal, and electric pulp tests, which were not considered here. Integrating these additional diagnostic measures could significantly enhance the precision and reliability of our findings, warranting further investigation in future research endeavors. The absence of histopathological examination to confirm PL diagnoses posed a significant limitation in our study, affecting both human observers and AI assessments. PRs were chosen due to their widespread use in dental imaging, offering broad anatomical coverage with relatively low radiation doses. We have to emphasize the importance of incorporating trans-hospital or hybrid datasets from diverse appliances and conditions for robust deep-learning applications. Our reliance on PRs from a single device in a single center further highlights a limitation warranting consideration in future research. In the present study, we achieved sensitivity (0.78), specificity (1.00), and diagnostic accuracy (0.89) values of a CNN-based AI tool for PL detection, which are in line with the reported findings of the available scientific literature. At this level, further scientific studies need to be conducted for a better understanding of how AI-based applications can support clinicians in the field of dental radiology in the future.

## 5. Conclusions

Based on diagnostic parameters, including the sensitivity, specificity, and diagnostic accuracy of DC, it can be concluded that the applied AI-based system supports the diagnosis of PL on PRs and serves as a decision-support tool during radiographic assessments.

## Figures and Tables

**Figure 1 diagnostics-15-00510-f001:**
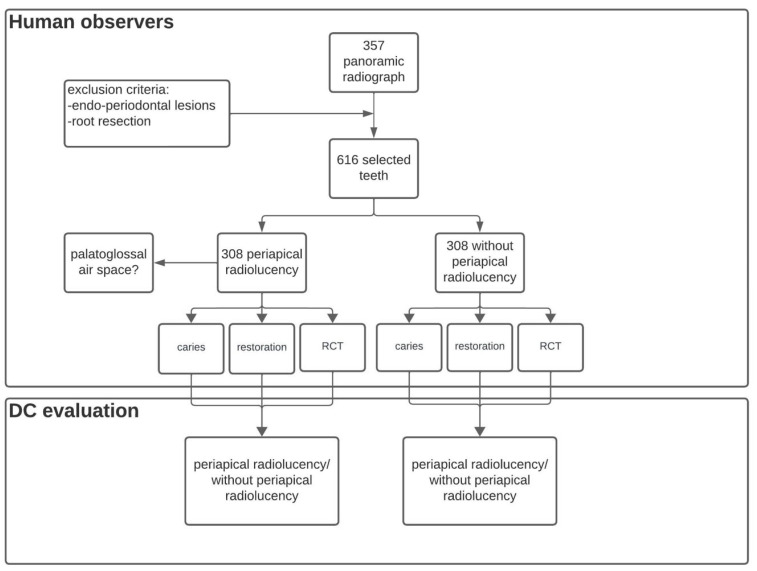
Data selection flowchart.

**Figure 2 diagnostics-15-00510-f002:**
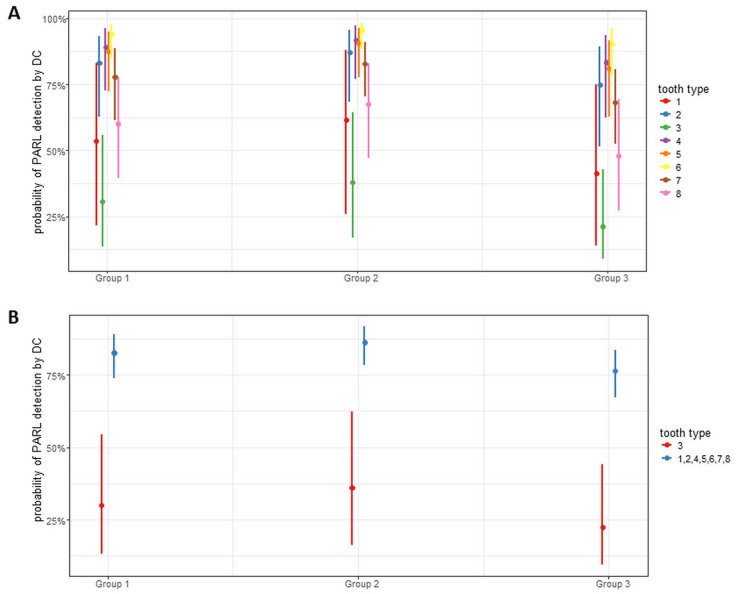
Plot diagram representing the results of logistic regression model for tooth type regardless of quadrants (**A**) and canines in relation to other teeth (**B**). Point estimate with 95% CIs of probability.

**Figure 3 diagnostics-15-00510-f003:**
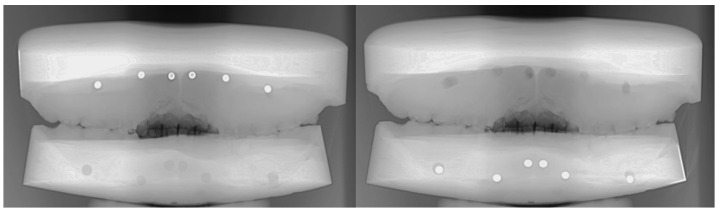
Panoramic radiographs of the prepared casts. The round opacities show the undistorted depiction of the embedded aluminum balls.

**Table 1 diagnostics-15-00510-t001:** Agreement of human observers and the DC.

	Human Observers		
DC	No PARL (*n* = 376)	PARL (*n* = 240)	TOTAL
no PARL	308 (100%)	68 (22.1%)	376 (61.0%)
PARL	0 (0%)	240 (77.9%)	240 (39.0%)

**Table 2 diagnostics-15-00510-t002:** Distribution of the three groups.

	No PARL (*n* = 376)	PARL (*n* = 240)	TOTAL
Group 1	70 (22.7%)	100 (32.5%)	170 (27.6%)
Group 2	228 (74.0%)	102 (33.1%)	330 (53.6%)
Group 3	10 (3.2%)	106 (34.4%)	116 (18.8%)

**Table 3 diagnostics-15-00510-t003:** The sensitivity, specificity, positive and negative predictive values, and correctly classified proportion values (95% confidence interval limits) of the DC overall and for each group.

	Overall	Group 1	Group 2	Group 3	Canines	Not Canines
sensitivity	0.78 (0.73, 0.82)	0.79 (0.70, 0.87)	0.84 (0.76, 0.91)	0.71 (0.61, 0.79)	0.27 (0.11, 0.50)	0.82 (0.77, 0.86)
specificity	1.00 (0.99, 1.00)	1.00 (0.95, 1.00)	1.00 (0.98, 1.00)	1.00 (0.69, 1.00)	1.00 (0.85, 1.00)	1.00 (0.99, 1.00)
positive predictive value	1.00 (0.98, 1.00)	1.00 (0.95, 1.00)	1.00 (0.96, 1.00)	1.00 (0.95, 1.00)	1.00 (0.54, 1.00)	1.00 (0.98, 1.00)
negative predictive value	0.82 (0.78, 0.86)	0.77 (0.67, 0.85)	0.93 (0.90, 0.96)	0.24 (0.12, 0.40)	0.58 (0.41, 0.74)	0.85 (0.80, 0.88)
a correctly classified proportion value	0.89 (0.86, 0.91)	0.88 (0.82, 0.92)	0.95 (0.92, 0.97)	0.73 (0.64, 0.81)	0.64 (0.48, 0.78)	0.91 (0.88, 0.93)

## Data Availability

The original contributions presented in this study are included in the article. Further inquiries can be directed to the corresponding author.

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
