# Peer review of "Deep Learning-Based Periapical Lesion Detection on Panoramic Radiographs"

_diagnostics, 2025, doi:10.3390/diagnostics15040510_

Round 1
Reviewer 1 Report
Comments and Suggestions for Authors
The study aimed to determine the accuracy of the artificial intelligence-based Diagnocat system (DC) in detecting periapical lesions (PL) on panoramic radiographs (PRs). However I have some concerns regarding the methodology and writing:
1. In line 58, the limitations of periapical radiographs, including the superimposition of anatomical structures, geometrical distortion, anatomical noise exist in the PRs, so I think it is not a valid argument to assert that PRs is more suitable as a tool for diagnosing periapical lesions. Please clarify the rationale for using periapical radiographs in a more logical manner.
2. Was the ground truth annotated by a senior dentomaxillofacial radiologist with over thirty years of experience, or by three independent observers? Please provide a more specific definition of the ground truth. Additionally, I believe that a senior radiologist and the dentomaxillofacial radiologist with more than ten years of experience, can accurately determine the ground truth. Why was it necessary for the other three observers to determine the ground truth?
3. Please explain in detail the reason for inserting the aluminum ball into the dental cast and subsequently performing a PR.
4. The table 1 should be divided into two separate tables because the content regarding the proportions of the three groups is independent of the content concerning the agreement between human observers and the DC. Combining these two sets of data into a single table is illogical.
5. In the part of discussion, the manuscript mainly compared the performance of the AI model used in this study with that of other AI models. I think that the differences in performance among the three groups and across different tooth types should be discussed and explained.
6. In term of the limitations mentioned in the discussion, why do the authors think a histopathological examination is necessary to confirm the PL?
Comments on the Quality of English Language
Can be improved.
Author Response
Comment 1: In line 58, the limitations of periapical radiographs, including the superimposition of anatomical structures, geometrical distortion, anatomical noise exist in the PRs, so I think it is not a valid argument to assert that PRs is more suitable as a tool for diagnosing periapical lesions. Please clarify the rationale for using periapical radiographs in a more logical manner.
Response 1: Thank you for your valuable comment. We acknowledge the limitations of periapical and panoramic radiographs, such as anatomical superimposition and geometrical distortion. However, PRs remain the most widely used imaging modality in general dental practice due to their relatively low radiation dose, cost-effectiveness, and ability to provide an overall assessment of the patient's dental status. Our study aimed to assess an AI-based system in a real-world clinical setting where PR are the most widely used modality. I included it in our manuscript in line 67-71.
Comment 2: Was the ground truth annotated by a senior dentomaxillofacial radiologist with over thirty years of experience, or by three independent observers? Please provide a more specific definition of the ground truth. Additionally, I believe that a senior radiologist and the dentomaxillofacial radiologist with more than ten years of experience, can accurately determine the ground truth. Why was it necessary for the other three observers to determine the ground truth?
Response 2: We appreciate your suggestion to provide a more specific definition of the ground truth. In our study, the ground truth was determined through a consensus-based approach. We changed it in line 111-114.
Comment 3: Please explain in detail the reason for inserting the aluminum ball into the dental cast and subsequently performing a PR.
Response 3: Thank you for your question. The aluminum balls were inserted into the dental cast to evaluate whether PRs introduce distortion or superimposition in the depiction of periapical areas. By placing these radiopaque markers at predefined depths corresponding to actual periapical locations, we aimed to simulate anatomical structures and assess whether radiographic projection affected the visibility of periapical lesions. The PRs of these casts confirmed that no significant distortion occurred, thereby validating the ability of PRs to depict periapical areas accurately in the study. You can read more about it from the manuscript from line 146 to 153.
Comment 4: The table 1 should be divided into two separate tables because the content regarding the proportions of the three groups is independent of the content concerning the agreement between human observers and the DC. Combining these two sets of data into a single table is illogical.
Response 4: We appreciate this suggestion and agree that dividing Table 1 into two distinct tables would enhance clarity. One table will present the distribution of the three groups, while another will focus on the agreement between human observers and the DC. We changed them in line 178-180.
Comment 5: In the part of discussion, the manuscript mainly compared the performance of the AI model used in this study with that of other AI models. I think that the differences in performance among the three groups and across different tooth types should be discussed and explained.
Response 5: Thank you for this recommendation. Notably, our results indicate that the AI system had lower sensitivity for detecting periapical lesions in canines compared to other tooth types. We hypothesize that this may be due to anatomical variations, such as the relatively long and narrow morphology of canine roots, which could affect lesion visibility. We expanded the discussion in line 276-281.
Comment 6: In term of the limitations mentioned in the discussion, why do the authors think a histopathological examination is necessary to confirm the PL?
Response 6: We acknowledge the reviewer's concern and will clarify our reasoning. Radiographic findings alone cannot distinguish between different periapical pathologies, such as granulomas, cysts, or abscesses, which may have similar radiographic appearances. Therefore, histopathological examination remains the gold standard for definitive diagnosis. However, since histological sampling is not feasible in routine clinical practice, AI-assisted radiographic assessment aims to enhance diagnostic accuracy while acknowledging this inherent limitation.
Reviewer 2 Report
Comments and Suggestions for Authors
The introduction fairly introduces the topic of the paper which is identifying periapical lesions from panaromic radiographs. The use of PR and its importance has also beein introduced fairly well
in methodology
were the human observers calibrated? if yes details of the calibration process should be included
line 132 - 159 discuss the methods used in the study and the challenges face, these should be place in the discussion section and not in the methods ideally
similarly lines 160-168 seem out of place and should be in the discussion
The discussion and conclusion are relevant to the study and are effective elaborated
Author Response
Comment 1: were the human observers calibrated? if yes details of the calibration process should be included
Response 1: Yes, thank you for this suggestion. Observers underwent calibration through a training session supervised by a senior dentomaxillofacial radiologist. They reviewed standardized radiographs, discussed discrepancies, and reached a consensus. We added these details to the Materials and Methods section from line 98 to 102.
Comment 2: line 132 - 159 discuss the methods used in the study and the challenges face, these should be place in the discussion section and not in the methods ideally
Response 2: Thank you for the suggestion! We have moved it to the Discussion section.
Comment 3: similarly lines 160-168 seem out of place and should be in the discussion
Response 3: Thank you for the suggestion! We heve moved it to the Discussion section.